# Vision-Language-Vision Auto-Encoder:
# Scalable Knowledge Distillation from Diffusion Models

**Tiezheng Zhang[1],   Yitong Li[2],   Yu-Cheng Chou[1],   Jieneng Chen[1],   Alan Yuille[1],**

**Chen Wei[3],     Junfei Xiao[1,†]**

[1]Johns Hopkins University    [2]Tsinghua University    [3]Rice University

[†]Project Lead

https://lambert-x.github.io/Vision-Language-Vision/

## Abstract

Building state-of-the-art Vision-Language Models (VLMs) with strong captioning capabilities typically necessitates training on billions of high-quality image-text pairs, requiring millions of GPU hours. This paper introduces the Vision-Language-Vision (**VLV**) auto-encoder framework, which strategically leverages key pretrained components: a vision encoder, the decoder of a Text-to-Image (T2I) diffusion model, and subsequently, a Large Language Model (LLM). Specifically, we establish an information bottleneck by regularizing the language representation space, achieved through freezing the pretrained T2I diffusion decoder. Our VLV pipeline effectively distills knowledge from the text-conditioned diffusion model using continuous embeddings, demonstrating comprehensive semantic understanding via high-quality reconstructions. Furthermore, by fine-tuning a pretrained LLM to decode the intermediate language representations into detailed descriptions, we construct a state-of-the-art (SoTA) captioner comparable to leading models like GPT-4o and Gemini 2.0 Flash. Our method demonstrates exceptional cost-efficiency and significantly reduces data requirements; by primarily utilizing single-modal images for training and maximizing the utility of existing pretrained models (image encoder, T2I diffusion model, and LLM), it circumvents the need for massive paired image-text datasets, keeping the total training expenditure under $1,000 USD.

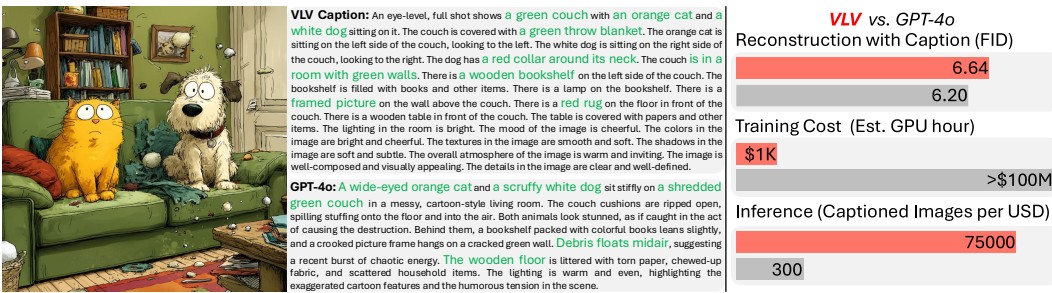

Figure 1: **VLV matches GPT-4o's descriptive fidelity at *three orders of magnitude* lower cost. Left:** VLV captures all salient objects, matching GPT-4o in coverage without hallucinations, yet better preserving their spatial layout. **Right:** On the FID–cost–throughput plane, VLV reaches comparable FID, trains for orders-of-magnitude less, and delivers vastly higher captions-per-dollar at inference—proving that detail-rich descriptions need not demand massive budgets.

# 1 Introduction

Multimodal representation learning aims to capture meaningful semantic relationships between vision and language. Broadly, existing approaches can be categorized into three major paradigms based on the interaction between textual and visual modalities: (1) vision-language models (VLMs), where images serve as inputs and text as outputs [50, 37, 5, 84, 68, 55, 52]; (2) contrastive learning frameworks [51, 83, 10], where image and text embeddings are aligned into a shared latent space through a contrastive objective; and (3) text-to-image generative models, where textual descriptions condition the generation of visual content [55, 52].

Traditionally, the first two paradigms, vision-language modeling and contrastive learning, have been predominantly utilized for learning robust multimodal embeddings. In contrast, text-to-image generative models, such as diffusion-based architectures [27], are generally considered generative tools rather than effective mechanisms for multimodal embedding learning. Although intuitively these generative models must implicitly encode detailed semantic relationships to produce coherent images, their potential for multimodal tasks like image captioning has not been fully realized.

Recent research suggests that text-to-image generative models indeed capture rich, nuanced semantic structures [66, 69], highlighting potential opportunities in applying the "analysis-by-synthesis" approach [82]. Rooted in cognitive science, this idea has long argued that perception works by imagining the hidden causes of a signal and selecting the one that best "explains" it. Motivated by this insight, our work demonstrates how pretrained text-to-image diffusion models can effectively transfer their inherently rich multimodal representations to downstream vision-language tasks such as captioning and VQA, where text-to-image diffusion models "imagine" the image, whose corresponding multimodal representation serves the "best explanation".

Specifically, we introduce a novel architecture termed the "**V**ision-**L**anguage-**V**ision" (VLV) autoencoder. In this framework, an open-source pretrained diffusion model, specifically Stable Diffusion 2.1 [55], is used as a powerful frozen diffusion decoder. We distill knowledge from this decoder into a bottleneck representation through a regularization process on the language embedding space produced by an encoder [72]. Next, these continuous intermediate representations are decoded through a pretrained LLM decoder [77] after alignment, generating detailed captions. Our approach achieves captioning performance competitive with leading proprietary models, including GPT-4o [1] and Gemini 2.0 Flash [22], while utilizing significantly smaller, open-source models.

Our methodology also exhibits strong scalability: We obtain substantial performance improvements when scaling the training dataset from 6M to 40M images. Notably, by primarily leveraging single-modal images, the data collection approach is much less of a burden compared to extensive paired image-text datasets. Adding up maximizing the utility of existing pretrained models, training costs remain below $1,000 USD (less than 1,000 GPU hours), significantly enhancing accessibility and promoting broader innovation within the vision-language research community.

Additionally, we explore emergent properties of the proposed VLV autoencoder: *a)* semantic richness, where learned embeddings encode detailed semantic aspects, including object 3D pose and orientation, resulting in robust spatial consistency; and *b)* compositional generalization, achieved by concatenating caption embeddings from distinct images, allowing the model to disentangle foreground objects from backgrounds effectively and compose novel, coherent, and visually plausible images.

In summary, the primary contributions of this work are as follows:

- We introduce **Vision-Language-Vision (VLV)** Auto-Encoder, a novel framework for scalable and efficient knowledge distillation from pretrained text-to-image diffusion models. This approach learns language-semantic representations only using image-based training.

- The construction of a lightweight yet effective LLM-based caption decoder, achieved by strategically integrating pretrained models, resulting in negligible training overhead.

- Comprehensive experimental results validate that the proposed captioner exhibits highly competitive captioning performance relative to SoTA VLMs, such as GPT-4o, and surpasses other open-source models of comparable parameter counts.

- An investigation into the emergent properties of the **VLV** framework, specifically highlighting the preservation of spatial semantics and advanced multi-image compositionality. These findings underscore the efficacy and potential of the learned representations.

## 2 Related Work

**Visual Autoencoder (VAE)** has long served as a foundational method for unsupervised representation learning [19, 24, 26, 32, 61]. Variants such as VQ-VAE [60, 54] and discrete VAE [53] extend this idea by learning discrete and structured representations. Although widely used for image tokenization in multimodal learning [21, 53, 55, 80, 81, 41], their quantized latent spaces often entangle semantics and require co-trained decoders to be effective. Recent works, like De-Diffusion [69], replace the latent bottleneck with text sequences decoded by pretrained diffusion models, aiming for interpretability. ViLex [66] further pushes this by directly training visual embeddings through generative supervision from frozen diffusion backbones, bypassing token-level representations entirely. Despite these closed-sourced advances, our VLV model is the first to efficiently build a vision-language-vision auto-encoder with all open-source modules with minimal training cost.

**Vision-Language Captioners.** Recent advances in vision-language models (VLMs) have significantly advanced image captioning by leveraging large-scale image-text pretraining and powerful multimodal architectures. Some previous works [79, 37, 36, 62, 73] aligned visual encoders with language decoders, while Flamingo [5], Kosmos [47, 29], and ShareGPT4V [11] highlighted few-shot and interleaved vision-text capabilities. Recent models like GPT-4o [2], Gemini [22], Qwen-VL [8, 76], and LLaVA [42] combined instruction tuning with powerful language backbones for fluent captioning. Large-scale systems such as PaLI-X [12], mPLUG-2 [74], InternVL [16], and CogVLM [63] scaled model and data size to achieve top performance on COCO [14], Flickr [48], NoCaps [3], and TextCaps [57], while IDEFICS [33], OpenFlamingo [6], Fuyu-8B [9], and Baichuan-omni [39] offered strong open-source alternatives. Emerging models like Emu3 [65], NVLM [17], Pixtral [4], and Molmo [18] further demonstrated the effectiveness of diverse multimodal modeling strategies. Despite these advances, most models depend on massive image-text pairs and costly training. In contrast, our VLV framework distills knowledge from a pretrained diffusion model using single-modal image data, enabling high-quality captioning without requiring web-scale, high-quality labels.

**Representation Learning with Diffusion Models.** A growing body of work has explored leveraging diffusion models for representation learning across diverse modalities and tasks [49, 67, 30, 59]. De-Diffusion [69] and ViLex [66] used frozen T2I models for language-aligned embedding learning. Other works, like DreamTeacher [35], distilled diffusion model features into discriminative backbones, while DiffMAE [70] recast denoising as masked autoencoding. Several studies also demonstrated that diffusion models can serve directly as zero-shot classifiers [34] or that their intermediate activations encode linearly separable features [71]. In the vision-language domain, SPAE [81] and RLEG [85] bridged image-language understanding using semantic autoencoding and synthetic contrastive supervision, respectively. ODISE [75] and DIVA [64] used diffusion priors to boost open-vocabulary segmentation and CLIP's perception, while RepFusion [78] explicitly mined time-step features for classification. Finally, simplification studies like Deconstructing DDMs [15] revealed that even stripped-down DAEs retain strong representational power. Unlike prior methods that require co-training of text and vision modules, handcrafted bottlenecks, or synthetic supervision, our method directly transfers generative knowledge into a latent space that supports both high-fidelity reconstruction and competitive caption generation with minimal compute.

## 3 Method

In this section, we introduce our proposed pipeline, which employs vision-language-vision (VLV) autoencoding to distill high-fidelity semantic information from images and subsequently decodes these semantics into descriptive captions using a multi-modal language model. We begin by outlining the pipeline architecture in §3.1. Next, in §3.2, we describe how we leverage a pretrained diffusion model to encode images into compact, continuous semantic embeddings, eliminating the need for explicit image-text pairs during training. Finally, in §3.3, we detail how these embeddings are decoded into natural-language captions via alignment with a pretrained large language model (LLM).

### 3.1 Pipeline Overview

VLV aims to extract high-fidelity semantic information from images through a pretrained T2I diffusion model. Previous similar work [69] utilizes discrete text token of CLIP as latent representation directly and Gumbel-Softmax [31, 45] for optimization, resulting in training inefficiency and lack of fine-grained semantic details. In contrast, we train our model using a continuous embedding space for

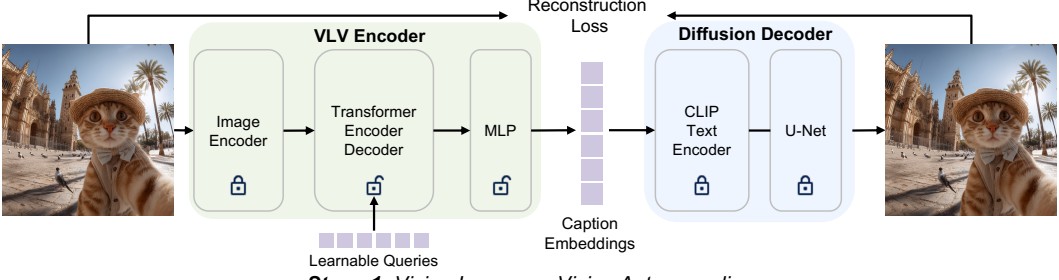

**Stage-1:** Vision-Language-Vision Auto-encoding

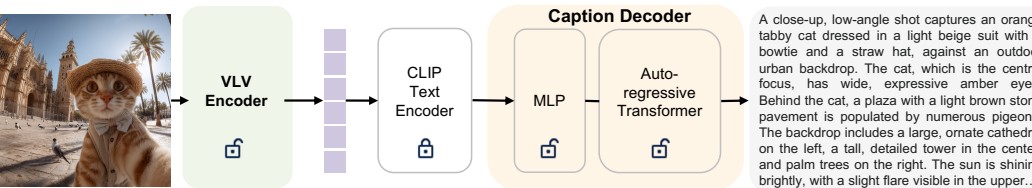

A close-up, low-angle shot captures an orange tabby cat dressed in a light beige suit with a bowtie and a straw hat, against an outdoor urban backdrop. The cat, which is the central focus, has wide, expressive amber eyes. Behind the cat, a plaza with a light brown stone pavement is populated by numerous pigeons. The backdrop includes a large, ornate cathedral on the left, a tall, detailed tower in the center, and palm trees on the right. The sun is shining brightly, with a slight flare visible in the upper...

**Stage-2:** Captioning with Auto-regressive Decoder

Figure 2: **Method Overview.** Our method has two stages: 1) vision-language-vision autoencoding for learning language semantics, 2) representation decoding into discrete language tokens through multi-modal LLM alignment. Our model has three major modules *(i) VLV Encoder*: a visual backbone augmented with a lightweight multi-modal adapter maps an input image into continuous caption embedding with compact semantic information; *(ii) Diffusion Decoder*: a *frozen* text-to-image diffusion model reconstructs the image; *(iii) Caption Decoder*: a pretrained large language model with an MLP projector decodes language-centric representations into comprehensive captions.

better training convergence, stability, and efficiency and decode the embeddings to discrete language tokens like multi-modal LLMs to generate text tokens given encoded visual embeddings of images.

Our VLV encoder extracts continuous *caption embeddings* directly from images. Training is fully self-supervised: a *frozen* text-to-image diffusion model serves as the decoder, reconstructing each image from its caption embeddings. Because the text-to-image diffusion model is fixed, the encoder must embed all information necessary for faithful reconstruction, effectively distilling the diffusion model's rich visual knowledge into a lightweight vision backbone, while eliminating the need for paired image–text data. Next, we fine-tune VLV encoder together with an LLM-based decoder that maps them to natural-language captions. Since the caption embeddings obtained by the VLV encoder are compact and encode only implicit semantics, we utilize a pretrained LLM to decode them into descriptive image captions. The autoregressive architecture of the LLM and its rich linguistic knowledge enable it to generate natural, coherent sentences with flexible length. This alignment uses paired image–text data specified in §4.1.

### 3.2 Knowledge Distillation from Diffusion Models

Following a self-supervised learning framework, this stage adopts a symmetric auto-encoder architecture that encodes to and decodes from latent tokens as information bottleneck. Given an image $x \in \mathbb{R}^{H \times W \times 3}$, a visual backbone produces visual tokens $v \in \mathbb{R}^{N_v \times D_v}$. A linear projection followed by LayerNorm [7] maps them to $v' \in \mathbb{R}^{N_v \times D}$. These tokens are concatenated with $N_t$ dummy prompt embeddings $t_{\text{prompt}}$ to form $X = [v'; t_{\text{prompt}}] \in \mathbb{R}^{(N_v + N_t) \times D}$, which a multimodal Transformer encoder converts to contextual states $h_E = \text{Enc}(X)$. Since there is no caption for supervision in this stage, we inject $N_q$ learnable query tokens $q \in \mathbb{R}^{N_q \times D}$ on the Transformer decoder side; cross-attention with $h_E$ yields $\hat{h} = \text{Dec}(q, h_E) \in \mathbb{R}^{N_q \times D}$. A lightweight MLP $\phi$ projects these states to the channel dimension of the frozen CLIP text encoder in the diffusion model, producing the *caption embedding* $z = \phi(\hat{h}) \in \mathbb{R}^{N_q \times d_{\text{CLIP}}}$. The text-to-image diffusion model $D$ remains *frozen*; it receives $z$ as conditioning and is optimised *only* indirectly. Specifically, with a latent $z_0 = E(x)$ and its noisy counterpart $z_t = \sqrt{\alpha_t} z_0 + \sqrt{1 - \alpha_t} \epsilon$, the frozen U-Net predicts the noise $\epsilon_\theta(z_t, t, z)$; the encoder parameters are updated by the standard denoising loss

$$\mathcal{L}_{\text{denoise}} = \mathbb{E}_{x,\epsilon,t} \big\| \epsilon - \epsilon_\theta(z_t, t, z) \big\|_2^2. \tag{1}$$

The auto-encoder architecture forces visual encoder to distill all information required for faithful reconstruction into the compact caption embedding $z$. Instead of using image-text paired data, visual encoder learns the inverse I2T mapping process through pretrained T2I diffusion decoder, which contains rich cross-modal knowledge. Rather than discrete text token and Gumbel-softmax, we use implicit and continuous embedding as latent for remaining detailed semantic information in a compact way without losing fidelity. The faithful encoding performed in Stage-1 forms the foundation for high-quality understanding and captioning in Stage-2, ultimately enabling accurate reconstruction.

### 3.3 Caption Decoding from Language-centric Representations

The aim of this stage is to decode intermediate representations into readable, high-quality captions. Previous structure design has fixed-length word tokens, contradicting with the inherent difference of complexities among all kinds of images, e.g., a picture of an apple and a picture of a big city should have semantic complexities of different levels. The setting limits the effectiveness and flexibility of image encoding, result in losing the potential of faithful reconstruction. Thus we introduce our LLM-based VLV Caption Decoder, which can decode unlimited and length-flexible natural language descriptions of images from compact semantic embeddings.

As shown in Section 3, we train our VLV encoder $E$ and LLM decoder $G$ with our image-text pair $(x, y)$. We first obtain the caption embeddings $z \in \mathbb{R}^{N_q \times d_{\text{CLIP}}}$ via VLV encoder (E). Since $z$ is in the CLIP text–embedding space, we pass it through the *frozen* CLIP text encoder $T$, obtaining contextual representations $c = T(z) \in \mathbb{R}^{N_q \times d_T}$. A lightweight trainable MLP $\psi : \mathbb{R}^{d_T} \to \mathbb{R}^{d_{\text{LM}}}$ then projects these vectors to the hidden size of a causal language model $G$: $e = \psi(c) \in \mathbb{R}^{N_q \times d_{\text{LM}}}$. During training with paired image–text pair $\{(x, y_{1:T})\}$, the projected vectors $e$ are *prepended* to the ordinary token embeddings of the caption, forming the input stream $[\, e;\, \text{Embed}(y_1), \ldots, \text{Embed}(y_T)\,]$. With positions corresponding to $e$ masked out, we compute the autoregressive loss only on real words:

$$\mathcal{L}_{\text{LM}} = -\sum_{t=1}^{T} \log p_\theta\big(y_t \mid e,\, y_{<t}\big), \tag{2}$$

where $\theta = \{E, \psi, G\}$ are the *only* trainable parameters; the CLIP text encoder $T$ remain untouched. At inference, we compute $z = E(x) \to c = T(z) \to e = \psi(c)$ and feed the projected vectors $e$ (without any text tokens) into the language model $G$, which autoregressively samples the caption. Thus this stage bridges the previous visual semantics to natural language with only a lightweight projection head, while fine-tuning $E$ and $G$ and keeping $T$ frozen. This design lets a compact latent embedding be flexibly decoded into human-readable captions of arbitrary length, while preserving fine-grained image semantics. And the progressive training-and-inference strategy achieves superior performance, as demonstrated empirically in Table 4.

## 4 Experiment

In this section, we first describe the experimental setup for both stages of VLV in § 4.1. Next, we report quantitative results on text-to-image (T2I) generation (§4.2.1), a human study of caption quality (§4.2.2), and visual-question-answering (VQA) benchmarks (§4.2.3). Finally, §4.3 presents two ablation studies: (i) a *trainable-parameter* study, varying the number of learnable queries for representation learning from information bottleneck and the progressive training strategy (i.e., progressively unfreezing encoder modules) in training the captioning decoder; and (ii) a *scalability* study in the aspects of training data scale and captioning decoder model size.

### 4.1 Experimental Setup

**Data Collection.** From LAION-2B-en-aesthetic, a subset of LAION-5B [56], we curate a 40M image subset. For training stability we keep only images whose shorter side is greater than 512, aspect ratio in the range of 0.5 to 2, and watermark probability less than 0.5. The resulting images are used to train the VLV auto-encoder under image-only supervision, without any accompanying text. Next, we query Gemini-2.0 Flash [58] to generate captions for 6M images in our dataset, producing aligned image-text pairs that fine-tune the lightweight language decoder. An overview for crafting our image-text pairs dataset used in alignment training is shown in appendix. Despite using only **0.4%** ($40M/10B$) of the WebLI dataset [13] used by De-Diffusion [69], our method still learns strong language-oriented semantics through the vision-language-vision auto-encoding pipeline.

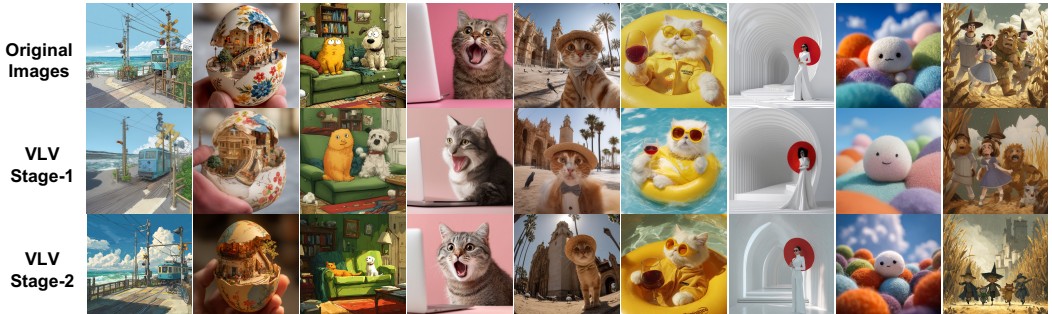

Original Images

VLV Stage-1

VLV Stage-2

Figure 3: **Reconstruction with language semantics.** For each original input image (top), we feed its *caption embedding* directly to the frozen diffusion decoder and obtain a reconstruction (middle) that preserves *high-level semantics and fine-grained appearance cues*. The same embedding is then decoded by the LLM; prompting Midjourney with that caption yields an image of high fidelity.

| Guidance Scale | 1.0 | 2.0 | 3.0 | 4.0 |
|---|---|---|---|---|
| Original [14] | 16.62 | 9.90 | 12.69 | 14.49 |
| Recap-7B [38] | 12.26 | 7.70 | 10.16 | 11.82 |
| LLaVA-v1.5-7B [42] | 14.54 | 9.99 | 12.93 | 14.91 |
| Qwen2.5-VL-7B [8] | 12.61 | 6.98 | 9.19 | 10.59 |
| Florence-2 Large [72] | 10.61* | 7.51 | 9.95 | 11.35 |
| **VLV (Ours)** | 11.47 | **6.64** | **8.56** | **9.90** |
| Gemini 2.0 Flash [58] | 12.82 | 5.87* | 7.57* | 8.77* |
| GPT-4o [2] | 12.16 | 6.20 | 7.96 | 9.25 |

Table 1: **Benchmark Captions Through Text-to-Image Reconstructions.** We evaluate the captions through FID scores ($\downarrow$), with image reconstruction. We use Stable Diffusion 3.5 Medium to reconstruct images with captions. Best results with open-source models are **bolded**; *: best for all.

| | Users | Gemini 2.0 | *Average* |
|---|---|---|---|
| Qwen2.5-7B [8] | 5.00 | 5.07 | 5.03 |
| GPT–4o [2] | 5.20 | 5.25 | 5.23 |
| **VLV (Ours)** | 5.17 | 5.18 | 5.18 |

Table 2: **Benchmark Captions Through Users and VLM Rating.** We asked human users and a state-of-the-art vision-language model (VLM), i.e., Gemini 2.0 Flash, to rate captions generated by different models, employing a scoring rubric ranging from 1 to 6. The evaluation criteria encompassed semantic accuracy, linguistic fluency, and relevance to the corresponding images.

**Training Details.** When training our VLV auto-encoder, we initialize the image encoder part with Florence-2 [72] pretrained weights. The additional $N_q = 77$ learnable queries are randomly initialized. We use AdamW [44] optimizer with $(\beta_1, \beta_2) = (0.9, 0.99)$ and a decoupled weight decay of $0.01$. Training runs for 200K steps with batch size 512 on 8 RTX™ 6000 Ada GPUs ($\sim 4$ days). The learning rate starts at 5e-5 and follows a cosine schedule [43]. We use Qwen-2.5 [77] pretrained models for initializing the LLM decoder. We train the captioning decoder with 100K steps, having the batch size of $64$. The learning rate decays linearly starting at 1e-5. We use FP32 in autoencoder training to make models converge with stability, while the LLM decoder training uses BF16.

## 4.2 Main Results

### 4.2.1 Text-Conditioned Reconstruction with Captions

We assess caption quality by feeding each decoded caption to *Stable Diffusion 3.5 Medium* [20] and computing the Fréchet Inception Distance (FID) [25] between the synthesized and original images on 30K samples from the MS-COCO 2014 validation split [14]. Captions are generated with four state-of-the-art VLMs: Florence-2 [72], Qwen2.5-VL [8], Gemini 2.0 Flash [58], and GPT-4o [2]. Image synthesis employs the *rectified flow-matching* sampler using 40 inference steps and classifier-free guidance [28] scale from 1.0 to 4.0. As Table 1 shows, our captions achieve an FID essentially indistinguishable from GPT-4o's (difference $< 0.5$) and markedly lower (better) than those of Florence-2 and Qwen2.5-VL, indicating that our captions convey visual semantics on par with the strongest public baseline; only the closed-source Gemini 2.0 Flash attains a marginally better score. Figure 3 shows qualitative results on generated images by both caption embeddings and corresponding decoded captions, illustrating the faithfulness of our caption embeddings.

### 4.2.2 Captioner Arena: Rating with VLMs and Humans

We benchmark caption fidelity by comparing state-of-the-art vision–language models (VLMs) with human raters under the identical three-criterion rubric—*coverage*, *no hallucination*, and *spatial-layout consistency*—and the 7-point rating scale (0–6) introduced in Appendix. A random sample

| Benchmark | Shot | Accuracy (%) | | | |
|---|---|---|---|---|---|
| | | Florence-2 Large | Qwen-2.5-VL-7B | Gemini 2.0 Flash | **VLV (ours)** |
| VQAv2 | 0 | 58.74 | 61.74 | 62.52 | 58.55 |
| | 4 | 60.05 | 62.37 | 63.36 | 61.72 |
| | 32 | 63.28 | 63.77 | 64.05* | 63.60 |
| OK-VQA | 0 | 46.80 | 45.83 | 46.34 | 45.31 |
| | 4 | 55.36 | 55.11 | 56.48 | 54.10 |
| | 32 | 59.72 | 61.20 | 62.31* | 60.25 |

Table 3: **Few-shot VQA Evaluation(Text-only).** We evaluate the VQA accuracy (%) on VQAv2 and OK-VQA under zero-shot or few-shot settings. DeepSeek-V3 answers *only* from the caption text. By 32-shot, VLV matches the best open-source model (Qwen-2.5) and sits within 1 percentage of the overall leader (Gemini 2.0 Flash), despite being far cheaper to train and run.

| $N_q$ | FID | Trainable modules | 1.0 | 2.0 | 3.0 | 4.0 |
|---|---|---|---|---|---|---|
| 16 | 5.72 | MLP only | 14.27 | 9.71 | 11.87 | 13.22 |
| 32 | 5.60 | MLP + LLM decoder | 12.22 | 7.55 | 9.58 | 10.84 |
| 77 | **5.30** | MLP+ LLM decoder + VLV encoder | 11.47 | **6.64** | 8.56 | 9.90 |

Table 4: **Ablation Studies.** Left: Effect of the number of learnable query tokens ($N_q$). Right: Effect of unfreezing modules in Stage-2; both reported by FID ($\downarrow$).

| # Images(M) | 1.0 | 2.0 | 3.0 | 4.0 | Decoder | 1.0 | 2.0 | 3.0 | 4.0 |
|---|---|---|---|---|---|---|---|---|---|
| 6 | 11.38 | 9.01 | 10.33 | 10.81 | Qwen-2.5 0.5B | 14.70 | 9.37 | 11.26 | 12.45 |
| 18 | 10.14 | 7.57 | 8.41 | 8.78 | Qwen-2.5 1.5B | 12.25 | 7.30 | 9.16 | 10.26 |
| 40 | 9.71 | **7.22** | 7.70 | 7.84 | Qwen-2.5 3B | 11.47 | **6.64** | 8.56 | 9.90 |

Table 5: **Scalability in Data and Decoder Scale.** FID ($\downarrow$) computed at guidance scales $1-4$ for (left) training-data size and (right) caption-decoder size. VLV demonstrates strong scalability.

of 200 images from the MS-COCO 2014 validation split [14] is paired with captions produced by Qwen-2.5 VL, GPT-4o, and VLV. Each image–caption pair is then evaluated by one VLM judge (Gemini 2.0 Flash) and three independent human raters. For every pair the judge returns a single score $s \in \{0, \ldots, 6\}$; the same rubric is applied by the human raters. Table 2 shows that VLV matches GPT-4o within $< 0.05$ points on the 0–6 scale, surpasses Qwen-2.5-VL-7B by $0.15$ on average, and is preferred by one of the three human raters. These results confirm that our caption embeddings yield human-level captions while remaining competitive with the strongest commercial VLMs.

### 4.2.3 Text-Only Question-Answering with Captions

Because our caption embeddings capture both global semantics and fine-grained appearance cues, we assess their effectiveness on open-ended vision–language tasks using VQAv2 [23] and OK-VQA [46] validation sets. Following Wei *et al.* [69], each caption is inserted as *image context* in a large-language-model (LLM) prompt, which the LLM then completes to answer the visual question. An answer is deemed correct only if it exactly matches the ground truth. We evaluate our captions with DeepSeek-V3 [40] in both zero-shot and few-shot settings, without any additional fine-tuning. Table 3 shows the zero-, 4-, and 32-shot accuracies using captions generated by different VLMs. In strict zero-shot, VLV trails the best baseline by roughly three percentage points, yet it gains the most from extra in-context examples (about five points on VQAv2 and fifteen on OK-VQA),so that by thirty-two shots it lies within a single point of the state of the art. Although VLV is not the top scorer in every setting, it reaches comparable while training at lower cost, underscoring its scalability.

### 4.3 Ablation Studies

We conduct two complementary ablation studies in this section. (1) **Trainable-parameter analysis.** We probe the impact of trainable parameters by (i) varying the dimensionality of the learnable queries when training VLV auto-encoder and (ii) selectively unfreezing individual modules of the VLV encoder while training the LLM decoder. (2) **Scalability analysis.** We test how performance scales by (i) scaling the training corpus from 6M to 18M and 40M images, and (ii) increasing the size of the autoregressive captioning decoder from 0.5 B to 1.5 B and 3 B parameters.

**Progressive Training Leads Better Performance.** Herein, we train VLV with different trainable parameters settings to explore the trade-off between performance and training cost. Stable Diffusion

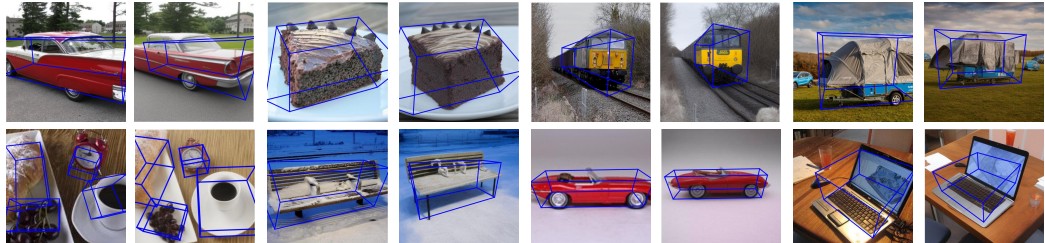

Figure 4: **Representation Learning Beyond Text: Spatial Preservation.** The figure compares the original images (left) with those reconstructed by our embeddings. The accurate 6D poses of individual objects and the relative spatial configurations among multiple objects demonstrate the method's strong capability in capturing spatial structure.

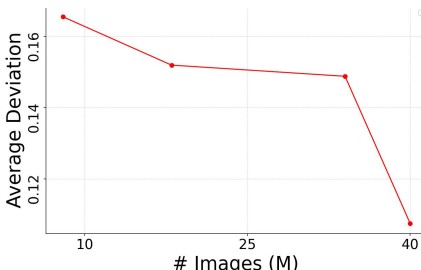

Figure 5: **Continual Spatial Representation Learning** VLV enables continual 3D spatial representation learning.

| # Images(M) | 8 | 18 | 34 | 40 |
|---|---|---|---|---|
| Angle (↓) | 0.1564 | 0.1287 | 0.1227 | **0.1016** |
| Center (↓) | 0.1625 | 0.1498 | 0.1402 | **0.0988** |
| Scale (↓) | 0.1775 | 0.1773 | 0.1835 | **0.1222** |

Table 6: **Quantitative Comparison of Spatial Awareness.** With more supervision images, VLV demonstrates improved spatial awareness. We evaluate this by measuring the L1 distance deviation between the bounding boxes of original and generated images with identical labels, as detected by Gemini 2.0 Flash [22].

2.1's CLIP text encoder accepts at most 77 tokens, and our default uses this full budget ($N_q$=77). We halve the number of learnable queries to $N_q$=16, 32 and gauge the impact by reconstructing MS-COCO 2017 test images from the resulting caption embeddings and reporting FID. In our second stage training, we progressively unfreeze the modules, starting with MLP first followed by the LLM decoder and finally the VLV encoder to see how many extra parameters are worth optimizing. Table 4 shows how reconstruction FID and caption quality improve smoothly with more trainable weights, clarifying the trade-off between performance and training cost.

**Scalability of VLV.** During training of the VLV auto-encoder we save intermediate checkpoints after the model has processed 6M and 18M images. To assess scalability, each checkpoint is used to extract caption embeddings for the 30 K images in the MS-COCO 2014 validation split described in §4.2.1. These embeddings are passed to the frozen diffusion decoder to reconstruct the images, and the resulting FID scores are reported in Table 5. We further probe model capacity by replacing the Qwen-2.5 3B caption decoder with its 1.5 B and 0.5 B variant while keeping all other components fixed (same table). In both cases FID degrades smoothly as data or decoder size is reduced, confirming that VLV benefits predictably from more training images and a larger language decoder.

## 4.4 Emerging Properties

### 4.4.1 Representation Learning beyond Text: 3D Visual Awareness

Besides rich details, we also find our embeddings have scalable spatial awareness. During training, as the diffusion decoder is exposed to a larger pool of images, the model steadily refines its spatial priors. To quantify this effect, we use Gemini 2.0 Flash to recover 3D bounding boxes for the primary objects in original images and compare them with boxes reconstructed from caption embeddings. Table 6 show a consistent reduction in pose estimation errors, and the examples in Figure 4 illustrate that VLV not only captures the poses of individual objects more accurately but also better preserves their spatial relationships. These results demonstrate that VLV effectively translates larger training image sets into sharper spatial understanding, as visualized in Figure 5.

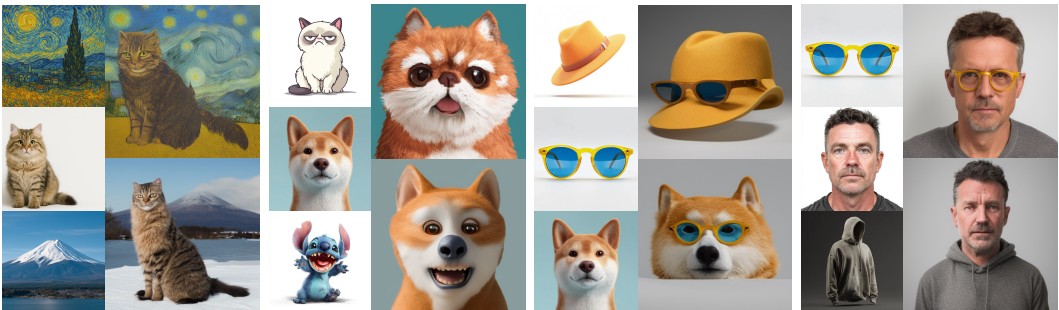

Figure 6: **Emerging compositionality with multi-image semantics.** Given two input images—a Siberian cat at the *left* edge of the frame and either (above) a Van Gogh-style painting or (bottom) a Mount Fuji landscape—we truncate and concatenate their caption embeddings and feed the composite vector to *Stable Diffusion 2.1*. The generated outputs faithfully preserve the cat's spatial layout while transferring the desired artistic style or background, *without any extra fine-tuning or text prompts*.

### 4.4.2 Compositionality with Multi-image Semantics

VLV semantic representation space exhibits strong *compositional* properties across multiple images, as illustrated in Figure 6. In the leftmost example, we begin with two images: (i) a photograph of a Siberian cat positioned on the *left* side of the frame, and (ii) a Van Gogh–style painting. By truncating the trailing tokens of each caption embedding and concatenating the resulting vectors, we create a joint embedding that is fed to *Stable Diffusion 2.1*. The synthesized output preserves the spatial layout of the cat while adopting the Van Gogh style, indicating that our embeddings encode both *content* (e.g., object identity and position) and *style* (e.g., artistic rendering). Notably, this compositional behavior emerges without any additional fine-tuning or reliance on text prompts. Further style transfer examples, including cartoon and Disney-style Shiba Inus, as well as try-on scenarios like a Shiba Inu or a man wearing sunglasses and a man trying on a hoodie or simple compositional of two objects like a Shiba Inu sitting in front of Fuji Mount and a sunglasses on a hat.

## 5    Conclusion

In this paper, we presented the Vision-Language-Vision (VLV) auto-encoder, a novel framework for scalable and efficient knowledge distillation from open-source pretrained text-conditioned diffusion models. By leveraging a strategically designed two-stage training process, VLV distills semantic-rich representations from frozen diffusion decoders into compact, continuous embeddings, and subsequently translates these embeddings into detailed natural language captions using an open-source pretrained Large Language Model. Our experiments demonstrate that VLV achieves state-of-the-art captioning performance comparable to leading models such as GPT-4o and Gemini 2.0 Flash, while dramatically reducing training costs and data requirements. Notably, our method primarily utilizes single-modal images, significantly enhancing accessibility by maintaining training expenditures under $1,000 USD. Additionally, we explored the emergent properties of our framework, highlighting its strong spatial consistency and advanced compositional generalization capabilities. We believe the efficiency, effectiveness, and interpretability of VLV pave promising pathways for future research in scalable and cost-effective multimodal learning.

**Limitations & Future Work.**    As our training data is filtered with aesthetic score, VLV performs poorly on OCR (Optical Character Recognition) tasks due to a lack of data with texts or watermarks; augmenting with document and street-view images or adding a lightweight OCR branch should somehow improve the performance on OCR scenarios. Another thing is that we are using the Stable Diffusion 2.1 as the generation decoder in our pipeline which is outdated also limits the transferable knowledge, limiting our upper bound. so re-distilling from recent state-of-the-art diffusion models such as SD 3.5 or FLUX is an incoming work. Moreover, extending VLV to video modality is also worthy to explore since videos offer more dynamics and could emerge stronger spatial representations as well as physics-based learning for understanding comprehensive world semantics.

## Acknowledgments

We thank the anonymous reviewers for their insightful comments. This work was supported in part by a grant from Amazon through the AI2AI program, and by the Office of Naval Research under award N000142412696.

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

# Appendices

# A Data Processing

This section details our data collection and filtering procedure. We annotate a subset of the corpus with *Gemini 2.0 Flash*[58]. Figure 7 shows the whole pipeline how we obtain our data for Stage-1 and Stage-2. Figure 8 provide the token length distribution of our captions used for training Stage-2.

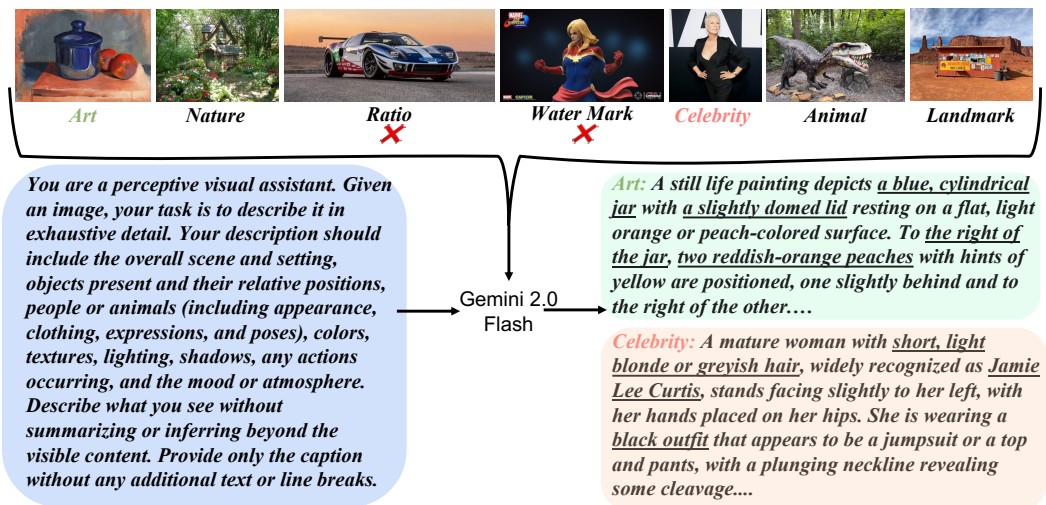

Figure 7: **Data Filtering Principles.** We filter and collect 40M images from LAION-2B-en-aesthetic. We apply filtering based on the image resolution and aspect ratio to ensure the image quality and then prompt Gemini 2.0 Flash with image-conditioned templates to generate rich, descriptive captions.

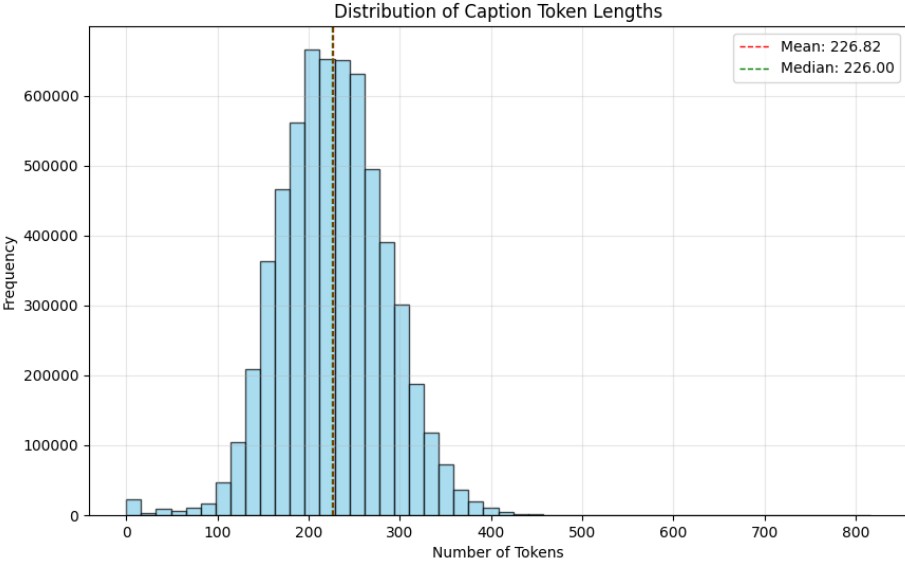

Figure 8: **VLV Captions' Length Statistics.** Histogram of token counts for all captions (our ∼6M image-text paired data, used for stage-2 captioning). Most captions fall in the $170-280$ token band, with mean $\mu = 226.82$ (red dashed) and median $\tilde{x} = 226$ (green dashed).

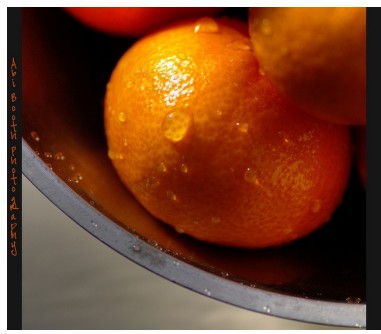

**Caption by VLV:** A close-up shot shows a bowl filled with three bright orange oranges, each glistening with water droplets. The bowl, which appears to be made of metal or a similar reflective material, has a dark, almost black interior. The oranges are positioned in the center of the bowl, with the one in the foreground taking up the majority of the frame. The other two oranges are partially visible in the background, slightly out of focus. The water droplets on the oranges create a sparkling effect, adding a sense of freshness and vibrancy to the image. The lighting is soft and diffused, casting gentle shadows and highlighting the textures of the oranges and the bowl. The overall mood is fresh and inviting, emphasizing the natural beauty of the fruit. The colors are vibrant and saturated, with the bright orange of the oranges contrasting against the dark background of the bowl. The composition is simple yet effective, drawing the viewer's attention to the oranges and their natural beauty. The focus is sharp on the oranges in the foreground, while the background is slightly blurred, creating a depth of field that draws the viewer's eye to the main subject. The image evokes a sense of simplicity and naturalness, highlighting the beauty of everyday objects. The overall impression is one of freshness, simplicity, and natural beauty.

Question: What states are these grown in?

Our Answer: Florida, California, Texas, Arizona

Gemini Answer: Florida, California

Ground-Truth:
["florida california", "florida california", "florida california", "florida california", "california", "california", "california", "california", "florida", "florida" ]

**Caption by Gemini:** A close-up captures several bright orange oranges with tiny water droplets clinging to their textured surfaces, situated in a silver or gray metallic bowl. The bowl's rim is visible along the bottom left quadrant of the image, showing slight discoloration and wetness from the condensation or water. The oranges are arranged in a clustered fashion, with one in the center and others partially visible around it, suggesting a group. The light is distributed unevenly, with highlights on the wet parts of the oranges and the bowl, and shadows cast on areas where they touch. The overall color palette is dominated by oranges, with the silver or gray bowl providing a neutral contrast. The texture appears rough due to the visible pores on the orange peels and the water droplets adding a reflective quality. The mood is fresh and inviting, potentially evoking a sense of healthy eating or refreshing citrus. On the left side of the photo is text, oriented vertically, in a thin, sans-serif font, that reads \"Abi Booth Photography.\"

Figure 9: **OK-VQA Example.** Both our caption and Gemini caption do not mention the states information. But our caption not only capture the oranges but also the number of oranges. Our answers contain the right ones highlighting in LimeGreen.

## B VQA Analysis: Are "Ground Truth" labels really ground truth?

Following Wei *et al.* [69], we evaluate on OK-VQA with DeepSeek-V3 [40] under the strict *exact-match* metric. Our raw score is $45.31\%$ (2,295 / 5,064), trailing the Gemini 2.0 Flash caption baseline of $46.34\%$ by $1.03\%$ (52 questions). Among the 526 cases where Gemini is marked correct and our model wrong, we compute answer–answer cosine similarity in CLIP space and relabel pairs with similarity $\geq 0.8$, recovering 94 additional correct answers. The adjusted accuracy is therefore $47.17\%$ This shows that the apparent deficit stems mainly from lexical mismatches rather than missing visual content. We show an example (one of the 94 cases) in Figure 9.

## C Vision-Language-Vision Autoencoding Does Help

We do an ablation study of the stage-1 Vision-Language-Vision autoencoding. To be specific, we only train our Stage-2 with pretrained our VLV Encoder, and assess the generated captions with T2I tasks. Table 7 reports the resulting FID scores (↓) on MS-COCO 2014. Skipping Stage-1 (first three rows) yields very poor fidelity, even larger decoders cannot compensate, whereas with Stage-1 training (grey row) drops FID to 12.2, confirming its critical role.

| Decoder | 1.0 | 2.0 | 3.0 | 4.0 |
|---|---|---|---|---|
| Qwen-2.5 0.5B | 57.29 | 64.02 | 66.92 | 68.18 |
| Qwen-2.5 1.5B | 49.15 | 56.66 | 59.15 | 60.14 |
| Qwen-2.5 3B | 45.63 | 51.03 | 53.28 | 54.40 |
| Qwen-2.5 3B + Stage-1 | 12.22 | 7.55 | 9.58 | 10.84 |

Table 7: **Effect of Stage-1 training on FID** (↓). The gray row demonstrates that our Vision-Language-Vision auto-encoding pipeline makes the encoder distill the knowledge from the text-conditioned diffusion model effectively and efficiently. This leads the effectively

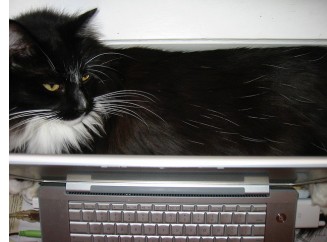

**Qwen-2.5 caption:** "The image shows a black and white cat lying on top of a laptop. The cat appears to be resting or lounging, with its body stretched out across the surface of the laptop. The laptop is open, displaying its keyboard, and the cat's head is positioned near the upper part of the laptop screen. The background seems to be indoors, possibly on a desk or table. The cat has a fluffy coat with a mix of black and white fur, and its eyes are partially closed, giving it a relaxed appearance."
**Rating:** 5.
**Explanation:** "The caption accurately describes the image, covering the main elements like the cat, laptop, and their relationship. It also mentions the cat's appearance and the indoor setting. There are no hallucinations."
**GPT-4o caption:** "A black and white cat with long fur and whiskers is lying stretched out inside an open laptop. The cat's body occupies the entire space where the screen would be, with its head resting on the left side of the laptop. Its ears are perked up, and it has an alert expression with its yellow eyes slightly narrowed. The keyboard of the laptop is visible in the foreground, featuring a silver color with white keys. The lighting is bright and even, casting soft shadows around the edges of the cat's fur. The texture of the cat's fur appears silky and smooth, while the laptop has a metallic and sleek appearance. The mood is lighthearted and quirky, suggesting a relaxed and playful use of the laptop as a resting place by the cat."
**Rating:** 6.
**Explanation:** "The caption accurately describes the key elements of the image, including the cat's appearance, position, and expression, as well as the laptop's features and overall setting. It also correctly identifies the spatial relationships between the objects."
**VLV caption:** "A close-up shot shows a black and white cat lying on top of a laptop, with the laptop's keyboard visible in the foreground. The cat's fur is a mix of black and white, with a prominent white patch around its face and chest. Its eyes are a striking yellow, and its whiskers are long and white. The cat's posture is relaxed, with its head turned slightly to the left. The laptop is silver and has a gray keyboard. The background is a plain white wall. The lighting is soft and diffused, creating subtle shadows on the cat's fur and the laptop's surface. The overall mood is calm and domestic. The focus is sharp on the cat and the laptop, while the background is slightly blurred. The texture of the cat's fur appears soft and fluffy, contrasting with the smooth surface of the laptop. The image is well-composed, with the cat positioned in the center of the frame, drawing the viewer's attention to its features and the laptop's keyboard."
**Rating:** 6.
**Explanation:** "The caption provides a comprehensive description of the image, covering the cat's appearance, the laptop, and the overall scene. It accurately describes the colors, textures, and spatial relationships between the objects. There are no hallucinations."

Figure 10: **Captioner Arena Example.** All captions show the correct objects without hallucinations. Both our caption and GPT-4o caption show the spatial relationship while Qwen-2.5 VL does not.

# D    Caption Evaluation with SoTA Multi-modal LLM (Gemini)

We assess caption quality by querying *Gemini 2.0 Flash* with a tailored rubric. Figure 10 displays an evaluation case with, together with Gemini 2.0 Flash's rationale, confirming that our captions are on par with those from GPT-4o.

---

Your role is to serve as an impartial and objective evaluator of an **image caption** generated by a Large Multimodal Model (LMM). Based on the single image input, assess the caption on *three* main criteria:

1. **Coverage of image elements** – how well the caption mentions the salient objects, their attributes, actions, and contextual details.

2. **Absence of hallucinations** – the caption must not invent objects, attributes, counts, spatial relations, or other details not present or implied by the image.

3. **Object spatial layout consistency** – whether spatial relationships (*left/right, above/below, front/behind, center, background/foreground*) are described accurately.

   - Any incorrect or invented spatial relation is a hallucination.
   - Omitting an obvious spatial relation reduces coverage.
   - Stating a relation that is ambiguous or uncertain in the image is also a hallucination.

**Evaluation protocol**
Start with a brief explanation of your evaluation process. Then assign *one* rating using the scale below. **Output only the rating number—no extra text, symbols, or commentary.**
6    Comprehensive coverage, correct spatial layout, *no* hallucinations
5    Very informative, correct spatial layout, no hallucinations, minor omissions
4    Moderate coverage, correct spatial layout, no hallucinations, several omissions
3    Limited coverage, minimal spatial detail, no hallucinations
2    Informative but contains at least *one* hallucination (object or spatial)
1    Limited coverage *and* at least one hallucination (object or spatial)
0    Not informative and/or multiple hallucinations

---

# E    Qualitative Results: Reconstruction from Captions

We show some qualitative results of our captions of MS-COCO 2014 validation split in Figure 11, Figure 12, Figure 13, Figure 14. In each figure, we show the original images and the reconstructed images generated by Text-to-Image generation models with our VLV captions. We show the generation results using Midjourney, FLUX.1-dev and Imagen 3. The reconstructed images preserve comprehensive semantics, demonstrating our VLV can do high-quality, comprehensive captioning.

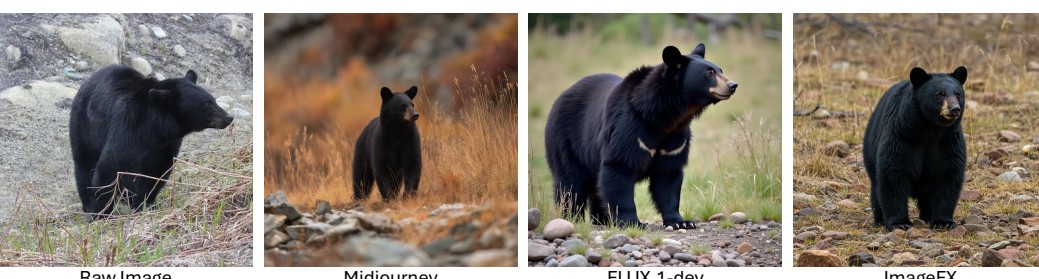

**Caption:** A medium shot captures a black bear standing in a grassy, rocky area. The bear is positioned in the center of the frame, facing slightly to the right, with its head turned towards the right side of the image. The bear's fur is a solid, glossy black, and its ears are small and rounded. It appears to be standing on its hind legs, with its front paws resting on the ground. The ground around the bear is covered with a mix of dry grasses and small, brown rocks. The rocks vary in size and color, ranging from light gray to dark brown. The grasses are a mix of green and brown, with some taller grasses reaching up towards the top of the frame. The lighting in the image appears to be soft and diffused, with no harsh shadows. The colors are muted and natural, with the black of the bear and the browns of the rocks and grasses dominating the scene. The overall mood of the image is peaceful and natural, capturing a moment of stillness in the bear's environment. The bear's posture and expression suggest a sense of calm and observation. The background is slightly blurred, drawing the viewer's attention to the bear in the foreground. The image is well-composed, with a clear focus on the bear and its surroundings. The texture of the bear's fur and the rough surface of the rocks are visible, adding depth and realism to the image.

Figure 11: **VLV can capture spatial layout.** The caption shows bear's layout *(in the center of the frame)* in this image as well as the bear's posture *(head turned towards the right side)*, showing VLV's ability of capturing spatial layout.

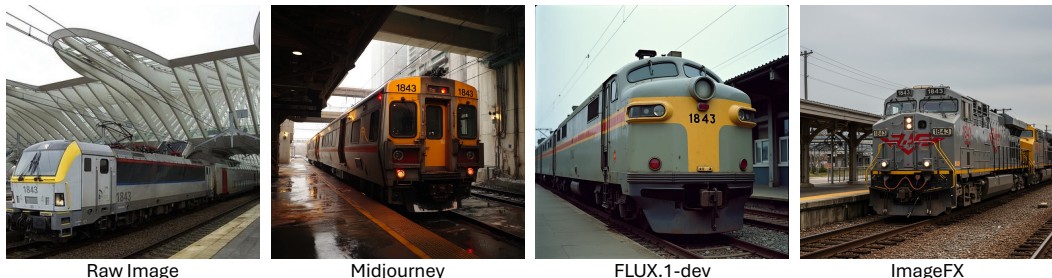

**Caption:** A medium, eye-level shot depicts a train car with the number \"1843\" painted on it, positioned on a train track in front of a train station. The train car is predominantly gray with yellow accents on the front and a red stripe running along the side. The number \"1843\" is prominently displayed on both the front and side of the car. The train track is visible beneath the train car, and the train station is visible in the background. The train station appears to be made of concrete and steel. The sky is overcast, and the lighting is soft and diffused. The overall mood of the image is calm and serene. The train car is the main subject of the image, and it is positioned in the foreground. The train station is visible in the background, and the sky is overcast. The lighting is soft and diffused, and the overall mood of the image is calm and serene.

Figure 12: **VLV can capture text (OCR).** VLV has reasonable OCR ability, even though the training set is heavily filtered (we filter the data by watermark probability less than 0.5). There is still potential to improve OCR performance with further training on more OCR-oriented data.

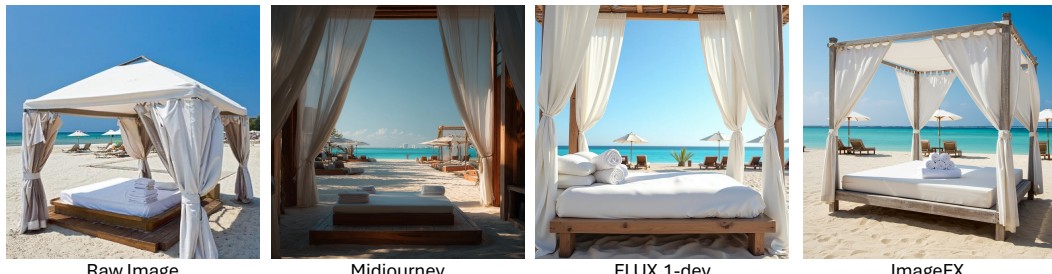

Raw Image    Midjourney    FLUX.1-dev    ImageFX

**Caption:** The image shows a luxurious outdoor bed setup on a sandy beach under a clear blue sky. The bed is elevated on a wooden platform and is covered with a white bedspread. A neatly folded stack of white towels sits on top of the bed. The bed is framed by white curtains that are tied back on either side, creating a sense of enclosure and privacy. The curtains are draped in a way that allows light to filter through, adding a soft glow to the scene. In the background, the turquoise ocean stretches out to the horizon, with several lounge chairs and beach umbrellas scattered along the shoreline. The sand is light-colored and appears to be well-maintained. The lighting is bright and sunny, casting soft shadows and highlighting the textures of the sand, wood, and fabric. The overall mood is serene and luxurious, evoking a sense of relaxation and vacation. The composition is well-balanced, with the bed as the focal point and the beach setting providing a natural and inviting backdrop. There are no people or animals visible in the image. The atmosphere is calm and peaceful, suggesting a perfect day at the beach. The colors are vibrant and natural, with the white of the bed and curtains contrasting against the blue of the sky and ocean. The textures are varied, from the smooth sand to the rough wood of the platform and the soft fabric of the curtains.

Figure 13: **VLV can capture complex objects.** Caption enumerates almost every object and correctly describe their spatial relationships, highlighting VLV's comprehensive scene understanding.

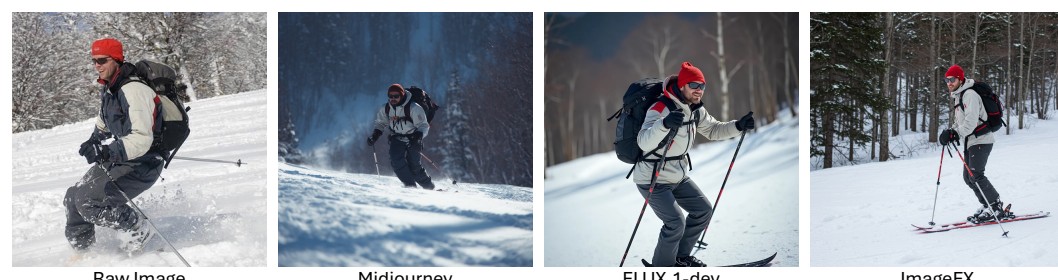

Raw Image    Midjourney    FLUX.1-dev    ImageFX

**Caption:** A medium shot captures a man skiing down a snow-covered slope. He is wearing a red beanie, dark sunglasses, and a light-colored ski jacket with black and red accents, paired with dark gray ski pants. A large black and gray backpack is slung over his shoulders. He is using ski poles to navigate the slope, and his expression is one of concentration and enjoyment. The background features a dense forest of bare trees, suggesting a winter setting. The ground is covered in snow, and the lighting is bright, casting shadows that add depth to the scene. The overall mood is one of outdoor adventure and winter sports. The texture of the snow is visible, adding to the realism of the image. The colors are muted, with the red of the beanie and jacket providing a pop of color against the predominantly white and gray tones of the scene. The man's ski tracks are visible in the snow, indicating his movement. The atmosphere is cold and crisp, typical of a winter day. The image is well-composed, with the man in the foreground and the forest in the background, creating a sense of depth. The lighting is bright and natural, enhancing the colors and textures of the scene. The overall impression is one of winter sports and outdoor adventure. The man's expression is one of enjoyment and concentration, adding to the overall mood of the image.

Figure 14: **VLV can capture human posture.** Captions show details of human as well as his posture, demonstrating VLV's fine-grained posture awareness.

# F    Dataset & Model License

## F.1    Training Datasets

**LAION-5B**

License: Creative Common CC-BY 4.0 https://laion.ai/blog/laion-5b/

## F.2    Testing Datasets

**MS-COCO**

License: Creative Common CC-BY 4.0 https://cocodataset.org/#termsofuse

**VQAv2**

License: CC-BY 4.0 https://visualqa.org/terms.html

Dataset website: https://visualqa.org/index.htmll

**OK-VQA**

License:N/A.

Dataset website: https://okvqa.allenai.org/

## F.3    Pre-trained Models

**stable-diffusion-2.1-base** (used for image generation).

https://huggingface.co/stabilityai/stable-diffusion-2/blob/main/
LICENSE-MODEL

**Qwen-2.5** (used in stage-2 for LLM decoder).

https://huggingface.co/Qwen/Qwen2.5-3B-Instruct/blob/main/LICENSE

**Qwen-2.5-VL** (used in image captioning).

https://github.com/QwenLM/Qwen2.5-VL/blob/main/LICENSE

**Florence-2-Large** (used in image captioning).

https://huggingface.co/microsoft/Florence-2-large/blob/main/LICENSE

**LLaVA-v1.5** (used in image captioning).

https://huggingface.co/liuhaotian/llava-v1.5-7b

