# OpenReview forum: "Vision‑Language‑Vision Auto‑Encoder: Scalable Knowledge Distillation from Diffusion Models"
_NeurIPS.cc/2025/Conference — NeurIPS 2025 poster_

### Official Review · Reviewer_CiwQ · 2025-06-16

**Clarity:** 2
**Significance:** 3
**Originality:** 3
**Rating:** 4
**Confidence:** 4

**Summary:**

This paper proposes a simple and low cost two-stage approach to generating good captions by distilling knowledge from t2I models.
The first stage is SSL based on images, where the VLV is required to learn good text embedding to align with the frozen t2I and reconstruct the input image. Based on the pretrained weights, the second stage utilizes text-image pairs and LLMs to construct a stronger captioner.
Some experimental results show  that this approach can achieve comparable captioning performance with GPT4o and Gemini 2.0 FLASH on some tasks.

**Questions:**

See the weakness part.

**Ethical Concerns:**

["NO or VERY MINOR ethics concerns only"]

**Final Justification:**

Thanks for the rebuttal. As most of my concerns have been addressed, I decide to raise my score.
It's better to include modern t2i models to make this paper stronger.

**Limitations:**

YES.

**Quality:**

2

**Strengths And Weaknesses:**

Strength
1. The paper studies an interesting topic which generates good text embedding for captions.
2. The image only pretraining design at stage 1 is novel, and I enjoy this idea.
3. The method is efficient and achieves comparable performance as GPT4o on some scenarios.

Weakness
1. During the training of stage 1, the diffusion decoder is frozen, how do you generate the image from the embedding? How many steps do you use to perform the de-noising process?
2. Section 4.2.1 and Table 1, how are the captions generated ?  What are the prompt templates for each VLM? How do you ensure a fair comparison, since VLMs are sensitive to the prompt template.
3. What's the validation set of Table~6? How many images are used to calculate the score? How do you predict the 3D bounding box?
4. Some detailed settings are missed. What's the configuration of Transformer encoder-decoder and MLP in VLV Encoder?
5. The evaluation is a bit limited and not comprehensive. A good caption model should cover many popular scenarios such as OCR and so on.
6. The component in the baseline is a bit old. It's more common to apply FLUX or SD3+.

Minors.
1. Line 198， For memory efficiency parameters are optimized using AdamW. Is there a direct link for this statement?
2. Line 205,   which do you use, FP16 or BF16? The statement of FP16/BF16 is confusing.

---

> ### Author Rebuttal · Authors · 2025-07-30
>
> 1. During Stage 1 of training, we follow the pipeline established by Stable Diffusion 2.1 base. Specifically, we first obtain conditioning features from the VLV encoder. These features are then passed through our trainable projection layer followed by the CLIP Text Encoder from SD 2.1. The resulting projected conditions, together with random noise, are input into the U-Net from SD 2.1 to predict the noise residuals. We employ a standard mean squared error (MSE) loss for supervision in this stage. Throughout training, we adopt the default 1000 denoising steps used in SD 2.1 for the noise prediction process. For inference (image generation), we follow the same conditioning pipeline and generate images using the SD 2.1 sampling procedure with 50 denoising steps.
>
> 2. It is indeed true that the VLMs are sensitive to the prompt template. Thus, for fair comparison, we use the same prompts for VLMs. Specifically, for `LLaVA-v1.5-7B` and `Qwen-2.5-VL-7B`, we use the same prompts. For `Florence-2 Large`, it has a fixed prompt template; for fair comparison, we use `<MORE_DETAILED_CAPTION>`. Please refer to the table below for detailed caption sources.
> | **Model**            | **Prompt / Caption Source**                                                                 |
> |----------------------|----------------------------------------------------------------------------------------------|
> | Original             | Captions from MS-COCO Dataset                                                               |
> | Recap-7B             | Captions from [1] .               |
> | LLaVA-v1.5-7B        | `Please describe this image as detailed as possible.`                                       |
> | Qwen-2.5-VL-7B       | `Please describe this image as detailed as possible.`                                       |
> | Florence-2 Large     | `<MORE_DETAILED_CAPTION>`                                                                   |
> | VLV                  | No prompt                                                                                   |
> *Table: Prompt templates or caption sources used for each model.*
>
> 3. For Table 6, we randomly select 200 images from *laion2B-en-aesthetic* as the validation set. And we prompt Gemini 2.0 Flash to predict the 3D bounding box for those images and then we visualize the boxes by ourselves. For Section 4.4, Emerging Properties,  the “emerging” aspect we highlight refers to the fact that our training pipeline unexpectedly yields images whose spatial structure can be reliably recovered in 3D—even though the model was never explicitly trained for that task.
>
> 4. Configuration of Transformer encoder-decoder:\
> The model employs a symmetric, sequence-to-sequence Transformer with 12 encoder and 12 decoder layers, each with a 1024-dimensional hidden size. Every layer uses 16-head self-attention, followed by a feed-forward network that expands to 4096 dimensions and applies a GELU activation. Regularization mirrors Stable Diffusion defaults: a 0.1 dropout is applied to activations, attention, and residual connections. The tokenizer supports sequences up to 4 096 tokens and uses the standard identifiers BOS = 0, PAD = 1, and EOS = 2. All weights are initialized from a normal distribution with σ = 0.02, ensuring stable training. \
> Configuration of MLP in VLV Encoder:\
> The MLP in the VLV Encoder is implemented as a two-layer feedforward network with a GELU activation in between. It projects language features from a 1024-dimensional input to a configurable hidden dimension (hidden_dim) and then back to 1024 dimensions.
>
> 5. We agree with the reviewer that handling OCR-related content is an important aspect for captioning models. As noted in our limitations section, our current model is not well-suited for OCR tasks. This limitation primarily arises from Stable Diffusion 2.1 itself. Stable Diffusion 2.1 is poor at generating images with plenty of words on those. Thus, we cannot distill the ability of capturing the OCR-related content from SD 2.1.
>
> 6. It is true that SD2.1 is a bit old; due to the computational resources, it is hard for us to apply SD3.5 and FLUX. Applying state-of-the-art diffusion decoders is also our future work.
>
> Minors:
> 1. We follow this setting from **De-Diffusion**[2] work and **Adafactor**[3].
> 2. Sorry for the confusion, we perform our experiment with both BF16 and FP16, and we find BF16 is the better option.
>
> Reference:\
> [1] **What If We Recaption Billions of Web Images with LLaMA-3?**
> [2] **De-Diffusion Makes Text a Strong Cross-Modal Interface**\
> [3] **Adafactor: Adaptive learning rates with sublinear memory cost**

---

> ### Comment · Reviewer_CiwQ · 2025-08-05
>
> Thanks for the rebuttal. As most of my concerns have been addressed, I decide to raise my score. It's better to include modern t2i models to make this paper stronger.

---

### Official Review · Reviewer_pgRD · 2025-07-01

**Clarity:** 3
**Significance:** 3
**Originality:** 2
**Rating:** 4
**Confidence:** 4

**Summary:**

this work proposes vision-language-vision autoencoders as a simple and cost-effective way to construct vision-language models by combining pretrained components including LLMs and text-to-image diffusion models. The training happens in two stages, where in the first one a frozen pretrained vision model is used to generate output tokens in the CLIP text encoder's space directly (ie the embedding layer after the tokenization), which are then used with a frozen t2i diffusion model to create an image. The connectors are trained with a simple self-supervised image-space reconstruction loss. In stage two, this model is used to train a simple captioner, ClipCap-style using image-text data.

**Questions:**

* Comparison to De-Diffusion on 40M images is missing. As a key comparison method that is both simpler and already public for a while, it would be important to comapre against this method.
 * what happens if you train the LLM with PEFT methods?
* how are 3D boxes constructed from the caption embeddings? this is not well described

**Ethical Concerns:**

["NO or VERY MINOR ethics concerns only"]

**Final Justification:**

I've read the rebuttal and the other reviews. My concerns have been addressed and I see the valuable contribution in terms of open-code and strong model performance, despite the conceptual incremental novelty in terms of method compared to De-Diffusion. I've increased my score.

**Limitations:**

yes

**Quality:**

3

**Strengths And Weaknesses:**

+ good idea of reusing pretrained components
+ simple training recipe and design choices
+ strong captioning performance

- Just a small point: "This approach learns language-semantic representations only using image-based training." might be confusing since you do rely on text-image supervised pretrained models.
- It seems that De-Diffusion is quite similar, except that they train discrete text tokens instead of continous ones, which further allows them to yield interpretrable captions directly, while VLV requires another stage for this.
- some experimental settings missing/unclear

---

> ### Author Rebuttal · Authors · 2025-07-30
>
> 1. To clarify our contribution: the core representation-learning stage relies exclusively on unpaired images, enabling the model to distill **language-semantic features** directly from visual data. Captioned text–image pairs are introduced only in a subsequent alignment phase to map the already-learned visual latent space onto the textual domain; they do not provide supervision for learning the underlying representations themselves.
> 2. De-Diffusion is **not open-sourced**, so a direct, fully controlled comparison is impossible. With their authors’ cooperation, we attempted to reproduce their pipeline; however, convergence proved unstable because their discrete-token approach relies on a Gumbel-Softmax relaxation whose temperature schedule is undocumented. In contrast, our method operates with continuous text embeddings, making the entire image-to-text pathway differentiable end-to-end and eliminating the need for potentially brittle discrete sampling. Moreover, we build on openly available pre-trained models and public datasets, ensuring that our codebase is reproducible for the research community and readily scalable in industrial settings. We believe these practical and methodological advantages provide a fair—and, for many applications, more deployable—alternative to De-Diffusion.
> 3. We added some experimental settings as **Reviewer CiwQ** requested, we hope those clarifications could help you have a better understanding of our work.
> 4. As requested, we have added a **LoRA-based baseline** to Stage 2. Specifically, we fine-tuned **Qwen-2.5 (3 B)** with **LoRA rank = 16** under exactly the same hyper-parameter settings used for the full-parameter baseline. After training, we generated captions for the same 30 K COCO validation images and assessed quality by prompting SD-3.5-medium and computing FID with guidance scale = 2. The LoRA variant reduces GPU memory consumption substantially (only the low-rank adapters are updated), but, as shown in Table here, this comes at a noticeable cost in caption quality, with FID increasing relative to full fine-tuning. These results confirm that while LoRA is attractive for resource-constrained scenarios, full fine-tuning remains preferable when maximum performance is required.
> | Method               | FID ↓ (GS = 2) | Peak GPU Memory (GB) per GPU |
> |----------------------|----------------|------------------------------|
> | LoRA (rank 16)       | 8.09           | 20 GB                        |
> | Full fine-tuning     | 6.64           | 47 GB                        |
>
>
> 5. Thank you for pointing out the ambiguity regarding the 3D bounding boxes in Section 4.4 (“Emerging Properties”). To clarify: the 3D bounding boxes are not produced directly from the text embeddings. Instead, after generating images from the text embeddings, we prompt Gemini 2.0 Flash to extract their 3D bounding boxes. The “emerging” aspect we highlight refers to the fact that our training pipeline unexpectedly yields images whose spatial structure can be reliably recovered in 3D—even though the model was never explicitly trained for that task. We will update the manuscript accordingly to make this workflow clear and avoid further confusion.

---

> > ### Comment · Reviewer_pgRD · 2025-08-05
> >
> > Dear authors, I've read the rebuttal and the other reviews. Great job on the rebuttal: my concerns have been addressed and I see the valuable contribution in terms of open-code and strong model performance, despite the perhaps conceptual incremental novelty in terms of method compared to De-Diffusion. I've increased my score.

---

> ### Comment · Area_Chair_RF3E · 2025-08-04
> **Please respond to the authors**
>
> Dear Reviewer pgRD,
>
> Thanks for your contribution to NeurIPS, and thanks for submitting your Mandatory Acknowledgement!
>
> Please note that the deadline of Author-Reviewer Discussions is fastly approaching (July 31 - Aug 6).
> Please acknowledge the authors' rebuttal as soon as possible.
>
> Thanks
>
> AC

---

> > ### Comment · Reviewer_pgRD · 2025-08-04
> > **I have already done that**
> >
> > Dear AC. I've already submitted my final rating and updated everything. Is it not visible?
> > Best.

---

> > > ### Comment · Area_Chair_RF3E · 2025-08-04
> > >
> > > Oh, it is only visible to chairs, not the authors. I think it would be better to acknowledge the authors as well.
> > >
> > > Thanks for your help!
> > >
> > > AC

---

### Official Review · Reviewer_hQYb · 2025-07-03

**Clarity:** 2
**Significance:** 3
**Originality:** 3
**Rating:** 4
**Confidence:** 4

**Summary:**

This paper introduces the Vision-Language-Vision Auto-Encoder, a two-stage framework for distilling cross-modal knowledge from pretrained text-to-image diffusion models into a lightweight image captioning system. In Stage 1, images are encoded into continuous “caption embeddings,” which a frozen diffusion decoder reconstructs back into images, effectively extracting semantic understanding without paired text labels. In Stage 2, these embeddings are fed into a frozen large language model that is lightly fine-tuned to generate natural-language captions. The authors demonstrate that, at a total training cost under $1,000, their method matches the caption quality of commercial systems like GPT-4o and Gemini 2.0 Flash on standard benchmarks.

**Questions:**

Please refer to the weaknesses.

**Ethical Concerns:**

["NO or VERY MINOR ethics concerns only"]

**Final Justification:**

Thanks to the authors for the rebuttal, which solved most of my problems. It would be more meaningful if it could be further verified based on other base models. I choose to maintain my rating.

**Limitations:**

Please refer to the weaknesses.

**Quality:**

3

**Strengths And Weaknesses:**

Strengths:
1. Achieves state-of-the-art captioning performance using only open-source models and modest compute budget, lowering the barrier for resource-constrained teams.

2. Cleverly leverages a frozen diffusion decoder as a semantic bottleneck, enabling self-supervised extraction of rich cross-modal features without paired supervision.

3. The continuous embeddings capture spatial layout and multi-item composition, enabling zero-shot reasoning about object relationships and 3D pose.

4. Fully based on public checkpoints with detailed training recipes, facilitating community adoption and future development.

Weaknesses:

Despite the paper’s interesting contributions, I have several serious concerns that need clarification:

1. The authors argue that pretrained diffusion models’ vision–language embeddings are highly valuable and therefore design a self-supervised pipeline to distill them. However, since CLIP’s text and image embeddings in these models are already well-aligned through extensive pretraining, why not simply extract the image embeddings directly? What concrete advantage does the proposed self-supervised scheme offer, if its main goal also seems to be alignment?

2. The current work uses Stable Diffusion 2.1, which relies on CLIP for text embeddings, but SD 3.0+ shifts to a T5-based text encoder. Does this mean the entire system must be retrained from scratch for newer SD versions? Or could one simply extract embeddings to achieve seamless compatibility across different frameworks? This point deserves a thorough discussion.

3. The authors employ DaViT as their vision encoder. Could alternative architectures (e.g., ViT variants, Swin, ConvNeXt) yield different or potentially better results within the same pipeline?

4. In Figure 3, some Stage 1 reconstructions differ in fine details—for example, whether a cat is sticking its tongue out. Might such minor discrepancies degrade the quality of downstream captions?

5. I’m curious whether the observed captioning improvements extend to more stylized or domain-specific imagery—such as posters, greeting cards, or other graphic art. Does the approach maintain its performance in these specialized scenarios?

---

> ### Author Rebuttal · Authors · 2025-07-30
>
> 1. The previous works, like LLaVA have already done this. And in our **Table1**, we have proved that our pipeline is better than previous ones. Moreover, compared with our pipeline, the CLIP requires **high-quality text-image pairs**. While in our pipeline, we can learn language semantics representation **via images only** in our training stage 1 by establishing an information bottleneck.
> 2. Retraining the entire system from scratch is unnecessary when moving to newer SD versions. In Stage 1, the VLV encoder is trained with an image-only, information-bottleneck objective and never encounters CLIP tokens. To accommodate a T5 text encoder, we simply introduce a lightweight linear/MLP adapter and fine-tune it. Concretely, we trained a two-layer MLP with a ReLU activation to map CLIP text embeddings to the T5 embedding space; after 2 000 steps on 512 k captions, the validation loss dropped to 0.003. Replacing the T5 encoder in SD 3.5 with the CLIP encoder from SD 2.1 plus this trained adapter still produced correct images. We use this pipeline for T2I tasks. Specifically, we generate hundreds of images with the corresponding captions and calculate the CLIP scores. We get the mean a CLIP score of **28.88**.\
> Because the text encoder in our pipeline remains frozen, functionally acting as a “large MLP layer”, adapting to any new text space only requires adding and briefly tuning a similarly small adapter.
> 3. To endow our model with genuine image understanding—i.e., the ability to map raw pixels to rich, language-semantic representations—we build the VLV encoder on DaViT, following Florence-2[1]. As Tables 7 and 9 in Florence-2 show(we put the table below for reference), DaViT consistently surpasses alternative backbones (ViT variants, Swin, ConvNeXt) on detection and segmentation benchmarks, indicating superior capacity for high-level visual reasoning. Leveraging this stronger visual core allows the VLV encoder to comprehend scene layout, object attributes, and relational cues more effectively. Consequently, substituting DaViT with other backbones would be unlikely to improve results within the same pipeline.
> ### Mask R-CNN / DINO Performance
>
> | Backbone        | Pretrain        | AP_b | AP_m | DINO AP |
> |-----------------|-----------------|:--------------:|:--------------:|:-------:|
> | ViT-B           | MAE IN-1k       | 51.6 | 45.9 | 55.0 |
> | Swin-B          | Sup IN-1k       | 50.2 | –    | 53.4 |
> | Swin-B          | SimMIM          | 52.3 | –    | –    |
> | FocalAtt-B      | Sup IN-1k       | 49.0 | 43.7 | –    |
> | FocalNet-B      | Sup IN-1k       | 49.8 | 44.1 | 54.4 |
> | ConvNeXt v1-B   | Sup IN-1k       | 50.3 | 44.9 | 52.6 |
> | ConvNeXt v2-B   | Sup IN-1k       | 51.0 | 45.6 | –    |
> | ConvNeXt v2-B   | FCMAE           | 52.9 | 46.6 | –    |
> | **DaViT-B**     | *Florence-2*    | **53.6** | **46.4** | **59.2** |
>
> ---
>
> ### ADE20K Semantic Segmentation (UperNet)
>
> | Backbone        | Pretrain        | mIoU | ms-mIoU |
> |-----------------|-----------------|:----:|:-------:|
> | ViT-B           | Sup IN-1k        | 47.4 | –    |
> | ViT-B           | MAE IN-1k        | 48.1 | –    |
> | ViT-B           | BEiT             | 53.6 | 54.1 |
> | ViT-B           | BEiTV2 IN-1k     | 53.1 | –    |
> | ViT-B           | BEiTV2 IN-22k    | 53.5 | –    |
> | Swin-B          | Sup IN-1k        | 48.1 | 49.7 |
> | Swin-B          | Sup IN-22k       | –    | 51.8 |
> | Swin-B          | SimMIM           | –    | 52.8 |
> | FocalAtt-B      | Sup IN-1k        | 49.0 | 50.5 |
> | FocalNet-B      | Sup IN-1k        | 50.5 | 51.4 |
> | ConvNeXt v1-B   | Sup IN-1k        | –    | 49.9 |
> | ConvNeXt v2-B   | Sup IN-1k        | – | 50.5   |
> | ConvNeXt v2-B   | FCMAE            | –    | 52.1 |
> | **DaViT-B**     | *Florence-2*     | **54.9** | **55.5** |
>
> 4. We acknowledge that feeding our caption embeddings into Stable Diffusion 2.1 cannot reproduce fine-grained visual details—the generation quality is capped by SD 2.1 itself. To mitigate this ceiling, Stage 2 trains an auto-regressive caption decoder jointly with a trainable VLV encoder. Making the visual encoder update alongside the decoder allows it to refine its representations toward the specific descriptive cues the decoder needs, yielding noticeably richer captions. The ablation in section 4.3 (Table 4) confirms the benefit: enabling VLV encoder updates yields the best performance among all other settings, despite the generative ceiling imposed by SD 2.1.
>
> 5. Yes. We use **laion2B-en-aesthetic** for stage 1 training to learn the language-semantic representation. This dataset is a subset of LAION-5B, which indeed contains stylized images, such as birthday greeting cards or posters. However, we filter out those images due to our criteria shown in the supplementary materials, Section A, data processing. We also explain that our model right now is not good at OCR due to the ability of SD 2.1. But our model could be applied to specific scenarios, fine-tuned on a corresponding dataset.
>
> Reference:\
> [1] **Florence-2: Advancing a Unified Representation for a Variety of Vision Tasks**

---

### Official Review · Reviewer_yw4J · 2025-07-03

**Clarity:** 4
**Significance:** 4
**Originality:** 4
**Rating:** 5
**Confidence:** 5

**Summary:**

This paper presents the Vision-Language-Vision (VLV) auto-encoder, a novel and efficient framework for leveraging pretrained text-to-image diffusion models to learn rich multimodal representations. The proposed two-stage pipeline first distills semantic information from images using a frozen diffusion decoder, then decodes the intermediate continuous embeddings into natural language captions with a pretrained large language model. Notably, the approach requires only single-modal image data and remains cost-efficient—under $1,000 USD for training—while achieving captioning quality comparable to state-of-the-art commercial models such as GPT-4o and Gemini 2.0 Flash. The authors further demonstrate that the learned representations exhibit strong spatial consistency and compositional generalization, underscoring the effectiveness of their distillation strategy.

**Questions:**

1) The authors state in lines 204–241 that “Although VLV is not the top scorer in every setting, it reaches comparable [performance] while training at lower cost, underscoring its scalability.” It would be helpful to clarify how cost-efficiency alone substantiates the claim of scalability. Typically, scalability refers to a model’s ability to handle increasing data or model size while maintaining or improving performance—how does lower training cost directly support this notion?

2) In Table 1 and Table 2, it is unclear which version of the Qwen2.5 language model was used (e.g., 1.5B, 3B, or larger such as 7B). While line 202 mentions Qwen2.5, it does not specify which model size corresponds to the key experimental results. Clarifying this would help contextualize the reported performance.

3) While the paper includes an ablation comparing 1.5B and 3B language decoders, it would be interesting to see whether the captioning performance continues to improve with larger models, such as 7B or beyond. Have the authors considered evaluating the proposed method with a more powerful LLM decoder to better characterize the scalability and upper-bound performance of the framework?

**Ethical Concerns:**

["NO or VERY MINOR ethics concerns only"]

**Final Justification:**

After reviewing the authors’ rebuttal, I have decided to maintain my original score of 5 (Accept). While some concerns remain partially addressed due to computational constraints, the core contributions of the paper are strong, and the authors have provided reasonable justifications and clarifications. My final assessment is based on the following points:
- Use of FID as a proxy for caption quality
The authors clarified that FID was chosen intentionally to evaluate detailed captions by measuring how well a diffusion model can reconstruct an image from the caption. While I still believe that including standard text-based metrics (e.g., CIDEr, SPICE) would strengthen the evaluation, their justification for relying on FID in the high-detail captioning context is reasonable. The acknowledgment of the limitations and discussion of trade-offs reflect thoughtful consideration of the evaluation methodology.
- Model scalability evaluation
Although the experiments do not include models beyond 3B due to GPU limitations, the authors do present a clear scaling trend from 0.5B → 1.5B → 3B, showing consistent improvements. This partially addresses my concern and supports their claim of scalability, even if not exhaustively demonstrated. I appreciate the authors’ transparency about resource limitations and their commitment to making the scaling trend more explicit in the final version.
- Modularity and diversity of pretrained components
The authors explained that using stronger diffusion models (e.g., FLUX, SD3.5) was infeasible within their current budget, but provided evidence that the current setup with SD2.1 is the limiting factor, not the captioner. This clarifies the rationale behind the fixed component choices and supports their claim of modularity under practical constraints.

Overall, while a broader set of evaluations would have further strengthened the paper, the authors’ thoughtful rebuttal and empirical evidence reinforce my positive impression of the work. The proposed framework is well-motivated, effective under realistic constraints, and makes a meaningful contribution to the field. I recommend acceptance.

**Limitations:**

Yes

**Quality:**

3

**Strengths And Weaknesses:**

# Strength

### Novel and well-motivated idea:
The core concept of using a frozen text-to-image diffusion model as a supervision signal to learn semantic-rich caption embeddings is original and intellectually engaging. The "Vision-Language-Vision" auto-encoding pipeline offers a fresh perspective on multimodal representation learning.

### Cost-efficiency:
The method demonstrates impressive performance despite using only single-modal images and a small compute budget (under $1,000 USD), making it accessible and reproducible for a wide range of research groups.

### Strong empirical results at small scale:
Even with relatively small-scale data (6M–40M images) and models (1.5B–3B), the approach delivers competitive or superior results compared to state-of-the-art systems, validating the effectiveness of the design.

### Scalability potential:
The observed scaling trends in both data size and model size suggest that the framework could further improve with larger-scale training. This indicates strong potential for future performance gains, making the method a promising foundation for more powerful captioning systems.


# Weakness

### Use of FID as an indirect and potentially confounded measure of caption quality:
The paper primarily assesses caption quality via FID scores computed from images synthesized by a third-party text-to-image model (SDM-3.5). However, this indirect evaluation conflates the quality of the generated captions with the capabilities of the generation model, making it difficult to isolate the effectiveness of the proposed captioner. Furthermore, standard automatic captioning metrics such as BLEU, METEOR, CIDEr, and SPICE are not reported, which limits comparability to existing captioning approaches. To better isolate the contribution of the captioner, the authors are encouraged to supplement their evaluation with standard automatic captioning metrics (e.g., CIDEr, SPICE, METEOR) on established benchmarks such as MS-COCO or NoCaps. These metrics would offer a more direct and model-independent measure of semantic and linguistic quality, improving the comparability and credibility of the reported results.

### Limited scope in scalability evaluation, especially in model scaling:
While the paper presents initial scalability experiments in terms of data (6M vs. 40M images) and model size (1.5B vs. 3B), the evaluation range remains limited—particularly on the model scaling side. Given that the proposed framework emphasizes cost-effective scalability, it would be beneficial to examine performance trends across a broader range of model sizes. In particular, extending the caption decoder to larger LLMs (e.g., 7B or beyond) could help reveal whether performance continues to improve with scale or begins to plateau. Although computational constraints are understandable, even an intermediate step toward a larger model would provide stronger empirical evidence for the framework’s scalability claims.

### Limited diversity of pretrained model components:
The proposed framework depends on a fixed set of pretrained modules—Stable Diffusion 2.1 as the frozen image decoder in Stage-1 and Qwen2.5 as the LLM decoder in Stage-2. While this setup is effective and cost-efficient, it leaves open the question of how well the method generalizes to more recent or stronger model architectures. For instance, replacing SD 2.1 with more capable T2I models such as FLUX, SANA, or Pixart-series may improve the quality of the learned semantic embeddings, and alternative LLMs like Mistral may offer gains in caption fluency or factual precision. Although the authors acknowledge this limitation for the image decoder, they do not provide empirical results with other models. If computational budget permits, evaluating the framework across a broader range of pretrained backbones would further strengthen the claim of modularity and demonstrate the robustness of the proposed approach.

---

> ### Author Rebuttal · Authors · 2025-07-30
>
> 1. We report FID rather than text-only metrics such as CIDEr because our task is detailed captioning: a caption is good if it contains enough information for a diffusion model to recreate an image that is perceptually close to the original. Concretely, we feed each caption to Stable Diffusion 3.5-Medium, which accepts prompts longer than 77 tokens, generate an image with identical denoising steps, resolution, guidance scale, and sampler across all experiments, and then compute the Fréchet Inception Distance between the generated and source images; since every setting except the caption is fixed, lower FID directly implies a richer, more accurate description. By contrast, CIDEr, BLEU, and SPICE depend on ground-truth references whose average length on MS-COCO is only about ten words, so they systematically penalise the longer, fine-grained sentences produced by modern VLMs. While FID thus provides a fairer gauge of descriptive adequacy in our scenario, we agree that the community still lacks a universally accepted metric for high-detail captioning.
> 2. Sorry for the confusion on the base model we use in our main experiments. We use Qwen2.5-3B as our LLM decoder in our training stage 2. Qwen 2.5-3B is the largest model we could afford given our GPU budget for fully fine-tuning.  But we do perform scalability experiments with Qwen2.5-0.5B, Qwen 2.5-1.5B, and Qwen2.5-3B. From the table here, it is obvious that with the increase of the LLM decoder size, the performance on the image captioning task improves. And according to the experiment here, we can infer that we can get better results with Qwen2.5 -7B or more powerful LLM.
> | Decoder    | GS=1 | GS=2 | GS=3 | GS=4 |
> |----------------------|---------------:|---------------:|---------------:|---------------:|
> | Qwen-2.5-0.5 B       | 14.70          | 9.37           | 11.26          | 12.45          |
> | Qwen-2.5-1.5 B       | 12.25          | 7.30           | 9.16           | 10.26          |
> | **Qwen-2.5-3 B**     | **11.47**      | **6.64**       | **8.56**       | **9.90**       |
> 3. Given our computational constraints, it was infeasible to adopt heavier diffusion back-bones such as SD-3.5, or FLUX for Stage 1, so we relied on Stable Diffusion 2.1. As illustrated in Fig. 3 (second row), SD-2.1 effectively defines our performance ceiling: when we reconstruct images directly from our caption embeddings, minor detail mismatches still appear, indicating that the decoder—not the caption model—is the limiting factor. To narrow this gap in Stage 2 we unfreeze the VLV encoder and train it jointly with the auto-regressive decoder; although this choice modestly increases memory usage, it yields clearly more faithful captions (Table 4). We therefore conclude that, resources permitting, pairing our pipeline with a more powerful diffusion model would further boost caption fidelity, while the current results already represent the best achievable quality under SD-2.1.
> 4. Thank you for pointing this out. Our intent in Lines 240–241 was not to claim that low absolute cost alone proves scalability, but rather to highlight that our pipeline already matches state-of-the-art performance despite running with modest-capacity diffusion and language decoders. The ablations in Table above show that when we increase only the LLM decoder from 0.5 B → 3 B parameters, caption quality (FID↓) improves monotonically. Thus, we expect better performance with a powerful diffusion decoder and LLM decoder. We will make this statement clearer in our manuscript once it is accepted.

---

> > ### Comment · Reviewer_yw4J · 2025-08-04
> > **Official comment**
> >
> > Since my main concerns are resolved, I would keep my initial acceptance rating.

---

### Decision · Program_Chairs · 2025-09-17

**Decision:**

Accept (poster)

**Comment:**

This paper introduces the Vision-Language-Vision (VLV) auto-encoder framework by combining pre-trained T2I diffusion model (DM) decoder and LLM. The paper proposes a two-stage pipeline where stage 1 is based on a distillation from DM (by decoding images from the intermediate feature), and stage 2 is for training a captioner from generated captions by LLM from the intermediate features. The proposed framework is efficient, showing training cost under 1,000 USD.

This paper has three positive borderlines and one positive recommendation.

The reviewers agree that the proposed framework is novel (yw4J, pgRD, CiwQ), efficient (yw4J, CiwQ), and effective even compared with large-scale systems (yw4J, hQYb, pgRD, CiwQ)

As unresolved weaknesses of the paper, the reviewers pointed out the problems in the limited set of scales and families of pre-trained models (yw4J, hQYb, CiwQ) and the problem of using FID as an evaluation metric for caption quality (yw4J). Also, pgRD pointed out that the novelty of the proposed method compared to De-Diffusion is somewhat incremental.

Even though the weaknesses still remain, the AC thinks that the weaknesses of this paper do not outweigh its strengths. I strongly suggest that the authors add more discussions related to the concerns raised by the reviewer.

Overall, I recommend accept this paper.